# Place, displacement, and health-seeking behaviour among the Ugandan Batwa: A qualitative study

**Laura Jane Brubacher**[1]*, **Lea Berrang-Ford**[2], **Sierra Nicole Clark**[3], **Kaitlin Patterson**[1], **Shuaib Lwasa**[4], **Didacus Namanya**[5,6], **Sabastian Twesigomwe**[7], **IHACC Research Team**[¶], **Sherilee L. Harper**[1,8]*

1 Department of Population Medicine, University of Guelph, Guelph, Ontario, Canada, 2 School of Earth and Environment, University of Leeds, Leeds, United Kingdom, 3 Environmental and Social Epidemiology Section, Population Health Research Institute, St. George's, University of London, London, United Kingdom, 4 Department of Geography, Makerere University, Kampala, Uganda, 5 Ugandan Ministry of Health, Kampala, Uganda, 6 Faculty of Health Sciences, Uganda Martyrs University, Kampala, Uganda, 7 Batwa Development Program, The Kellermann Foundation, Kanungu District, Uganda, 8 School of Public Health, University of Alberta, Edmonton, Alberta, Canada

¶ Membership of the Indigenous Health Adaptation to Climate Change (IHACC) Research Team is provided in the Acknowledgments.
* ljbrubacher@uwaterloo.ca (LJB); sherilee@ualberta.ca (SLH)

**Data Availability Statement:** Data is available in the supporting information files.

**Funding:** Financial support was provided by CIHR/NSERC/SSHRC and IDRC Tri-Council Initiative on

## Abstract

For many Indigenous Peoples, relationships to the land are inherent in identity and culture, and to all facets of health and wellbeing, physically, emotionally, psychologically, and spiritually. The Batwa are Indigenous Peoples of rural, southwest Uganda who have experienced tremendous social and economic upheaval, due to relatively recent forced displacement and land dispossession. This loss of physical connection to their ancestral lands has significantly impacted Batwa health, and also affected available healthcare options for Batwa. This exploratory study (1) identified and characterized factors that influence Batwa health-seeking behaviour, using acute gastrointestinal illness, a critical public health issue, as a focal point for analysis; and (2) explored possible intersections between the Batwa's connection to place—and displacement—and their health-seeking behaviour for acute gastrointestinal illness. Twenty focus group discussions, stratified by gender, were conducted in ten Batwa settlements in Kanungu District, Uganda and eleven semi-structured interviews were conducted with primary healthcare workers, community health coordinators, clinical officers, and development program coordinators. Qualitative data were thematically analyzed using a constant comparative method. Batwa identified several significant motivators to engage with Indigenous and/or biomedical forms of healthcare, including transition to life outside the forest and their reflections on health in the forest; 'intellectual access' to care and generational knowledge-sharing on the use of Indigenous medicines; and Batwa identity and way of life. These nuanced explanations for health-seeking behaviour underscore the significance of place—and displacement—to Batwa health and wellbeing, and its relationship to their health-seeking behaviour for acute gastrointestinal illness. As such, the results of this study can be used to inform healthcare practice and policy and support the development of

Adaptation to Climate Change, Indigenous Health Adaptation to Climate Change (IHACC), IDRC File nos. 106372-003, 004, 005 (IHACC); CIHR Open Operating Grant, Adaptation to health effects of climate change among Indigenous Peoples in the global south (IP-ADAPT), Application no. 298312 (IHACC), and a CIHR Vanier Canada Graduate Scholarship (LJB). The funders had no role in study design, data collection and analysis, decision to publish, or preparation of the manuscript.

**Competing interests:** The authors have declared that no competing interests exist.

a culturally- and contextually-appropriate healthcare system, as well as to reduce the burden of acute gastrointestinal illness among Batwa.

## Introduction

For many Indigenous Peoples, relationships to the land are inherent in the formation of personal and cultural identity, and to all facets of health and wellbeing, physically, emotionally, psychologically, and spiritually [1–3]. Indeed, place attachment [4, 5], sense of place, and place-based identity are central in many Indigenous epistemologies and ontologies–ways of knowing and understanding reality [1, 6]. As Cummins posits, "[place] *constitutes* as well as *contains* social relations and physical resources" [7:1825]; as such, place is a significant determinant of health [7, 8]. Health-seeking behaviour, then, emerges from, and is influenced by, broader social norms, attitudes, values, and behaviours unique to a population [9–12]. Considering that many Indigenous Peoples live within distinct socio-cultural, political-economic, and environmental contexts, place may play a significant part in Indigenous Peoples' health-seeking behaviour. While research has shown that place affects health-seeking behaviour through place-mediated factors such as resource availability and neighbourhood socioeconomic status [13–16], less research has examined the influence of place on health-seeking behaviour when Indigenous Peoples experience displacement from their ancestral lands.

The Batwa (Twa) are Indigenous Peoples of rural, southwest Uganda who have experienced tremendous social and economic upheaval, due to relatively recent forced displacement and land dispossession [17]. In 1991, the Batwa were evicted–with little to no compensation or recourse–from their ancestral forest lands due to the establishment of Bwindi Impenetrable Forest National Park [18, 19]. Batwa must adhere to rules strictly enforced by the Ugandan Wildlife Authority to enter the park. As such, access to the forest remains restricted for most Batwa [20]. The impetus and rationale for this conservation initiative, supported by the World Bank/Global Environment Facility Trust Fund, was to ensure protection of a declining mountain gorilla population. This initiative had, and continues to have, wide-ranging repercussions for Batwa, whose livelihoods and sense of place are tied to the forest [19, 21]. Batwa were forced to transition from a semi-nomadic, subsistence, hunting and gathering livelihood in the forest to an agricultural livelihood and wage economy. This overhaul of the Batwa way of life eliminated access to Indigenous forest foods, medicines, and shelter; changed the forms of healthcare available to Batwa; and, consequently, led to discrimination and social, economic, and political exclusion [21–26].

The heterogeneity and pluralism of Batwa health-seeking behaviour in response to illness, including the use of different forms of healthcare such as Indigenous and biomedical, has been discussed elsewhere [27]. Moreover, physical and economic access to healthcare, the efficacy of biomedical and Indigenous forms of healthcare, and the quality of biomedical healthcare, are all reported to influence Batwa health-seeking behaviour to varying degrees [27]. But, given the importance of place to Batwa identities and livelihoods, critical gaps remain in understanding whether or not Batwa connection to place, and their displacement from ancestral lands, may influence health-seeking behaviour; and if so, how displacement may shape their engagement with healthcare services.

In order to explore the dynamics between place, displacement, and health-seeking behaviour for Batwa, this study examined experiences with acute gastrointestinal illness, involving diarrhea and/or vomiting [28]. Acute gastrointestinal illness is the fifth leading cause of death

worldwide and, thus, a critical public health issue [29]; however, a gap exists in acute gastrointestinal illness research with Indigenous populations [30, 31], who often experience a high prevalence of infectious illnesses and low levels of biomedical healthcare access in response [32–35]. For Batwa, acute gastrointestinal illness has been self-identified as a key health priority [22]. Recent studies have reported a high burden of acute gastrointestinal illness among Batwa [30], particularly in children under three years of age (14-day prevalence = 11.3%), but also diverse health-seeking behaviour in response [27]. Understanding health-seeking behaviour and individuals' perceptions of local healthcare systems is critical to the development of healthcare infrastructure that reflects Indigenous values, place-connections, and conceptualizations of wellbeing, and, ultimately, supports the health of Indigenous Peoples [36–38].

The objectives of this study, then, were to (1) continue to identify and characterize the factors that influence acute gastrointestinal illness-related health-seeking behaviour for Batwa in Uganda; and to (2) explore possible intersections between the Batwa's connection to place–and displacement from their ancestral forest territory–and their health-seeking behaviour for acute gastrointestinal illness. Improved understanding of the impact of displacement on the health, wellbeing, and health-seeking behaviour of Batwa is not only relevant to their healthcare context, but also to forced displacement and resettlement of Indigenous Peoples worldwide, and the diverse healthcare systems that may work to support them in culturally-safe ways.

## Methods

### Ethics statement

All research processes were approved by the Research Ethics Boards at McGill University (REB File #486–0514) and the University of Guelph (REB File #14MR002). Informed verbal consent was provided by all participants (as many participants did not read or write) and documented in a secure log sheet. Verbal consent processes were approved by the above Research Ethics Boards.

### Partner communities and research approach

There are ten Batwa communities in the Kanungu District of southwestern Uganda: Bikuuto, Buhoma, Byumba, Karehe, Kebiremu, Kihembe, Kitahurira, Kitariro, Mukongoro, and Rulangara. Kanungu district includes a total of 47 health facilities (including private and government-funded) that vary in the level of care they offer, from small healthcare centres with primary healthcare capacities to two hospitals that provide specialized treatments and surgeries: (1) Kambuga Hospital, a federally-funded and -managed facility with free services for patients; and (2) Bwindi Community Hospital, a privately-funded facility. At Bwindi Community Hospital, Batwa are able to access subsidized health services and prescription medications through the *eQuality Health Bwindi* insurance program provided by the Batwa Development Program. This program is an initiative of the Kellermann Foundation non-profit organization (https://www.kellermannfoundation.org).

This study was guided by a participatory, community-based research approach [39, 40]. The research design was inherently collaborative, consisting of individuals representative of Batwa communities, governmental and non-governmental organizations, as well as academic researchers from Uganda, Canada, and the UK with experience in community-led Indigenous health research (Fig 1). The research focus emerged from Batwa priorities [22], and Batwa engaged in and co-led data collection and validation processes. Additional information regarding the ethical, cultural, and scientific considerations specific to inclusivity in global research is included in the Supporting Information (S1 Checklist).

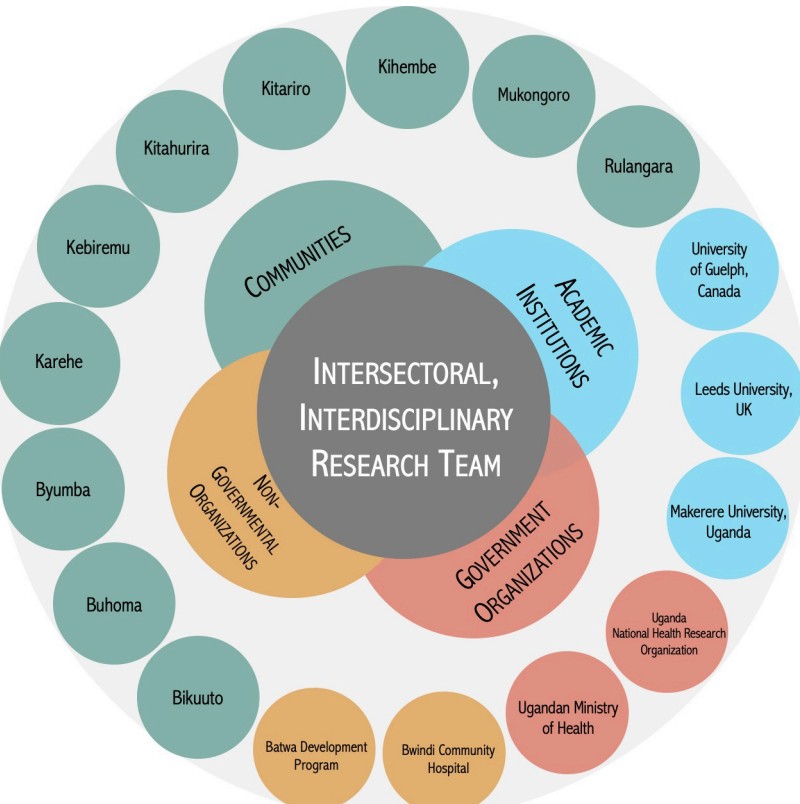

**Fig 1. Visual representation of the intersectoral, interdisciplinary research team involved in this study.** Green = 10 partner communities in Kanungu District; Blue = Academic institutions in Canada, the United Kingdom, and Uganda; Orange = National-level, governmental organizations; Yellow = District-level, non-governmental stakeholders, including the Batwa Development Program of the U.S.-based non-profit organization, The Kellermann Foundation.

## Data collection and analysis

From June 1 to July 31, 2014, qualitative data were collected as part of a larger study on the burden of acute gastrointestinal illness in Kanungu District, Uganda [30, 31, 41] (see discussion guides in S1 and S2 Files). Twenty focus group discussions were conducted in ten Batwa settlements. Focus groups were stratified by gender to better reflect local norms related to gathering and knowledge-sharing. Discussions were held in community gathering spaces within each settlement. Sociodemographic data collected as part of the larger study [30] indicated low socioeconomic variation among households from the ten settlements and typically no education or primary-level education among households.

Additionally, eleven semi-structured interviews with individuals working in healthcare and for governmental and non-governmental organizations were conducted at locations that were private and convenient for participants (e.g. place of employment). One video interview was conducted via Skype (Fig 2). All data gathering procedures were audio-recorded, with permission. Interviews were conducted in English, while focus group discussions were conducted in the local language of Rukiga. Transcription of the audio-recordings was conducted in partnership with a local translator. This individual was also an interviewer and, therefore, better able to subsequently review transcripts for accuracy of interpretation [42, 43].

Thematic analysis of these data proceeded iteratively and consisted of initial open coding and reflective memoing to generate possible codes [44, 45], followed by a constant comparative

## QUALITATIVE DATA GATHERING

### Focus Group Discussions (n=20)

**2 groups per community:**
- 1 female group
- 1 male group

| Bikuuto | Buhoma |
| Byumba | Karehe |

| Mukongoro | Kebiremu | Kihembe |
| Rulangara | Kitahurira | Kitariro |

### Semi-Structured Interviews (n=11)

- Total time: 272 minutes
- Average interview: 27 minutes
- Range: 20-38 minutes

**8 male, 3 female:**
- Primary healthcare workers
- Clinical officers
- Community health coordinators
- Individuals affiliated with national & international non-profit organizations in Kanungu District & Uganda at-large

**Fig 2. Qualitative data gathering procedures used in this study.** These procedures include twenty gender-stratified focus group discussions with ten Batwa communities and eleven semi-structured interviews with a variety of individuals in healthcare, as well as governmental and non-governmental organizations.

approach to collapse codes into themes present within and between interview and focus group discussion transcripts [46]. A codebook was developed using Dedoose, then codes were applied to each paragraph of text in the transcripts [46]. Preliminary results were reviewed with representatives from the Batwa communities; this form of member checking was conducted as a validation procedure, to help ensure that key themes were accurately interpreted and that Batwa knowledge and values were appropriately reflected and represented in the analysis [47].

## Results

### Transition from 'place' and reflections on health in the forest

In interviews, many Batwa individuals reflected on their acute gastrointestinal illness symptoms when they lived in the forest and their health-seeking behaviour in response to those symptoms. As stated by one interviewee:

> In the forest we had really good health because we would access the herbs that would treat any illness. But, we are failing to get used to the food that we found people eating when we got out of the forest, because we believed that any food we ate while in the forest was medicine (male participant).

Batwa individuals explained that acute gastrointestinal illness was *"common today and not when we were in the forest"*, because *"in the forest people would access body strengthening foods: meat, honey, and some herb. . .that would keep them strong and prevent them from getting several diseases, including diarrhea, but today. . .we are no longer eating the food that we are supposed to eat"* (female participants). This significance of forest foods to overall strength and illness prevention was reiterated by many participants:

In fact, the food we used to have in the forest was good and would make us strong. We never used to fall sick while in the forest. We can't find that kind of food again because we are not allowed to go back to the forest (female participant).

Some participants' reflections on health in the forest were presented in contrast to their current state, in which *"we are always weak"* (female participant). As a Batwa participant shared, *"Our way of life has changed, because we can't live like we lived in the forest"* (female participant). Interviewees, who work closely with the Batwa in hospitals and development organizations, also discussed the Batwa's transition from the forest, as *"they did not get a chance to be told how to cope with the new life they were leading,"* and that *"the Batwa were uprooted from their culture and thrown into a new experience which they were not prepared for"* (male participant). This transition from place was described as an experience of loss by an interviewee: *"In Africa, land is everything. It is everything, and when the Batwa lost their land they lost their ancestral home, their livelihoods, their heritage and their home"* (female participant). As evidenced by these results, the Batwa's displacement has had ongoing implications for their health and health-seeking behaviour (see S3 File for additional exemplar quotations for each theme).

## 'Intellectual access' to care

Some participants' comments suggested that decreased 'intellectual access' to healthcare also influenced Batwa health-seeking behaviour. Many described constraints to sharing or using knowledge of Indigenous medicines, as they are denied physical access to the forest. As one Batwa participant explained,

If they had not stopped us from going back to the forest, we would be freely accessing the herbs and teaching our children how to use them, as well as giving the herbs to them for treatment when they are sick (male participant).

Comments like these were associated with broader discussions of generational knowledge-sharing, how it has been affected by forest eviction, and how this influenced their health-seeking behaviour. Focus group participants shared that *"the new generation [of Batwa children] will not know where to get the herbs from"* (male participant), and that *"the new generation will keep suffering from this disease because they never tasted the strengthening foods that the Elders would eat when they were in the forest. . .we can no longer go to the forests because we are not allowed. . .we tell [children] to go to the hospital only"* (male participant). Further, one focus group participant expressed the transience of herbs as treatment, that *"modern medication might lead to the use of local herbs disappearing"* (female participant). A number of similar comments reflected this transience:

Use of local herbs is disappearing already and we might find our children not [using] them so much in the future. Local herbs were most used when we were in the forest by our grandparents, but since we moved away from the forest, we no longer use them (female participant).

Other participants discussed not knowing where to find herbs, since Batwa can no longer access their ancestral territory, and now rely on finding Indigenous herbs near their agrarian plots; however, when shown photos of herbs in the interviews, many Batwa readily identified herbs by name and described their methods of preparing the herbs as acute gastrointestinal illness treatment. This knowledge of Indigenous medicines appeared to differ from participants' knowledge of biomedicines for acute gastrointestinal illness. For instance, as a participant

stated, *"I just take the drugs [from the hospital] but don't know what they are called"* (male participant). Batwa expressed less interest in acquiring knowledge of biomedicines, stating: *"We can't bother knowing the names for the drugs, for as long as we take them, we get cured"* (female participant). Being denied forest access, the less-known locations of Indigenous herbs outside the forest, and the increased use of biomedicines for acute gastrointestinal illness by some Batwa, were reported constraints to sharing knowledge of Indigenous medicines with future generations. As participants explained, this decreased 'intellectual access' to Indigenous healthcare has potential to shift the health-seeking behaviour of future Batwa.

## Batwa identity and way of life

Connected to the Batwa's comments on Indigenous knowledge and 'intellectual access' to Indigenous or biomedical healthcare, is the concept of identity. Some participants alluded to their health-seeking behaviour also being a function of cultural identity and of the historical continuity of Batwa having used herbs for generations. As one Batwa participant explained, *"we can't stop using the local herbs even if we can access hospitals because **traditionally we are meant to use the herbs** [emphasis added]"* (male participant). A participant also noted that *"every herb is medicine to the Batwa"* (female participant). Similarly, one participant suggested that their decision to not seek any care was influenced by cultural identity: *"I just didn't attend to him and left him. . .until he got healed, you know **the Batwa way of life** [emphasis added]"* (male participant). Other participants suggested that self-sufficiency and independence as Batwa also influenced the form of healthcare they chose: *"We take local herbs and after that we get healed because that's what we used to take even before we had hospitals–and we would get healed"* (female participant). Furthermore, the socio-cultural and cosmological beliefs upheld by Batwa, as elements of collective identity, also influenced their health-seeking behaviour. As a participant explained, *"our ancestors are saying our lack of access to the forest is making us sick"* (female participant). An interviewee further identified that *"some people don't believe that germs are responsible for the diseases. They think it is a familial spirit or something like that, so they may not bring the patient in and may try some traditional method of treatment"* (male participant). As shared, some Batwa's beliefs as to the causes of acute gastrointestinal illness, which can be socially- and culturally-constructed, also influenced their resultant health-seeking behaviour. Individual participants' choice of acute gastrointestinal illness healthcare was reportedly shaped, then, by their collective identity as Batwa.

## Discussion

This study provides insight into the role of place and displacement in Batwa health-seeking behaviour. Batwa participants shared nuanced explanations for health-seeking behaviour that underscore the significance of place–and displacement–to Batwa health and wellbeing, and its relationship to their health-seeking behaviour for acute gastrointestinal illness (Fig 3). As such, the results of this study may be used by healthcare providers and policy-makers to develop a culturally- and contextually-appropriate and supportive healthcare system [7].

### Transition from 'place' and reflections on health in the forest

Batwa participants' reflections on their transition from place and health experiences in their ancestral forest territory reflect an emerging literature on centrality of place-connections to health and wellbeing, particularly that of Indigenous Peoples [6, 48–52]. Research suggests that how individuals or communities conceptualize and practice wellbeing can be place-specific and relevant to their particular environmental and cultural contexts [7, 48, 53]. In this light, the Batwa's displacement may have continual and long-term health and wellbeing

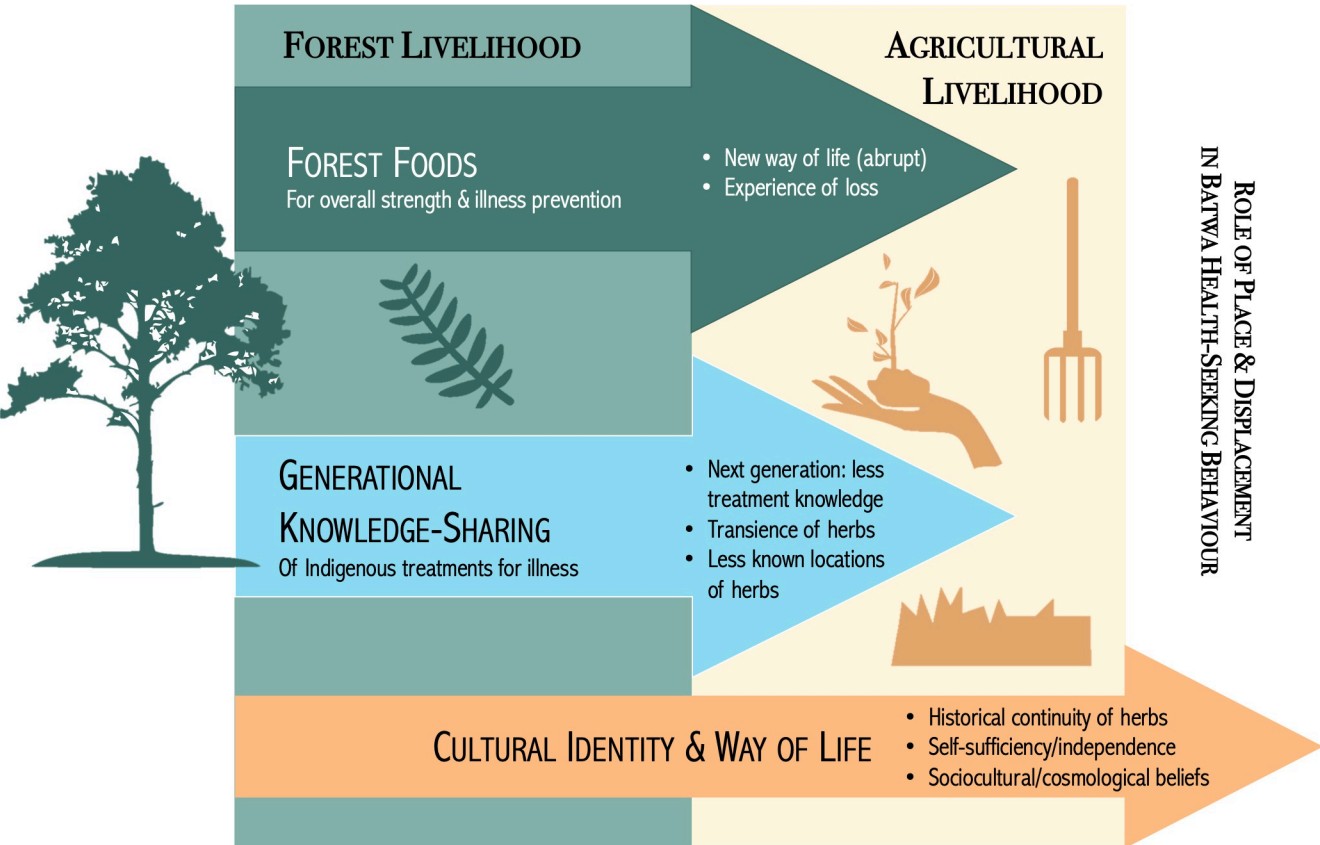

**Fig 3. Visual synthesis of results from this research, indicating the role of place and displacement in Batwa health-seeking behaviour for acute gastrointestinal illness, as shared by participants.** The transition from a forest-based, hunting and gathering livelihood to an agricultural livelihood impacted availability of foods important for health and illness prevention; generational knowledge-sharing of Indigenous treatments for acute gastrointestinal illness; and Batwa cultural identity and way of life tied to the use of Indigenous treatment.

implications. For many Indigenous Peoples, health is inherently and inextricably linked to relationships: kinship relationships with the land and one another, sustained through practices that occur on and with the land in particular places [6, 50]. Displacement necessarily alters relationships to the land, or at least constrains how Peoples can relate to their lands in the ways they always have. This research suggests, then, that the Batwa's health-seeking behaviour may be unavoidably affected, and probes the question: *How does one 'seek health' when the context in which health and wellbeing are regularly understood and practiced has dramatically changed*?

Studies conducted with internally-displaced northern Ugandans [49] and Cambodians [54], and sub-Saharan African refugees in Australia [55], suggest that health-seeking behaviour at an individual and group level is influenced by displacement in a number of distinct ways: Displacement can change social norms or cultural values [49]; impede awareness of the forms of healthcare available in one's context [54]; disrupt social structures and social capital (such as a family or village unit) that influenced the types of healthcare people once used [54]; create barriers to healthcare access, such as transportation or language barriers [55]; and lead to "cultural bereavement" and homesickness that undermine one's emotional capacity to seek healthcare [55:595]. Many of these impacts may similarly be felt by Batwa and influence their health-seeking behaviour [27]. Moreover, linkages between the Batwa's displacement and health-seeking behaviour may be further shaped by their Indigenous, place-based epistemologies and ontologies [1, 6, 48]. Further research might explore linkages between place attachment and

the health-seeking behaviour of Indigenous Peoples, in particular, as a means of informing how public health initiatives, health policy, and primary healthcare practice can more fully support Indigenous Peoples in engaging forms of healthcare they desire–Indigenous, biomedical, none, both, or yet other forms [27].

## Batwa identity and way of life, and 'intellectual access' to care

Batwa discussed how a sense of cultural identity, and identity emergent from socio-cultural worldviews and place, informed their health-seeking behaviour. King et al. (2009) describe many Indigenous identities as collective and thus intrinsically linked to social identities [56]– the 'self' as in relation to family, community, the land, and animals, as "ecocentric" [57:292]. For Batwa, then, denial of forest access may fundamentally represent some loss of identity that was formed by relationship to their ancestral lands, viewed as "a loss of their rights to practice and protect their entire epistemology" [20:33]. Forced into a new place, this loss of, or change to, a community's social fabric–integrally land-based–can greatly affect the health-seeking behaviour of displaced peoples [54].

Similarly, 'intellectual access' to care relates to the Batwa's place-based knowledge system, created, sustained, and understood within their forest lands [19, 20]; this extends from Batwa hunting and gathering livelihoods, to their knowledge of herbs and other Indigenous medicines to treat acute gastrointestinal illness. Outside the forest, Batwa reported not knowing how to access Indigenous forms of healthcare, a barrier to practicing Indigenous medicine that has been reported elsewhere [58]. Similar to Batwa health-seeking behaviour, other studies identify a stronger sense of cultural identity and an historical experience of loss as factors associated with more frequent use of Indigenous healing practices [58–60]. The concept of time is also implicit within each explanation of Batwa health-seeking behaviour presented in the results: For instance, an interviewee suggested that, in time, the use of herbs may be replaced by biomedical healthcare for acute gastrointestinal illness, while others discussed a possible loss of Indigenous treatment knowledge with time. Health-seeking behaviour as a function of time, and the temporality of Indigenous healthcare systems, has been echoed in pharmacological studies in Uganda, too [61].

As Batwa continue to transition from their ancestral forest to a new agrarian reality, their knowledge of herbal acute gastrointestinal illness treatments may, too, be adaptable to this new environment. Batwa need be recognized as highly-knowledgeable in forest ecology and conservation, expertise that can be utilized in their new context [62]. These results on Batwa identity and Indigenous treatment knowledge suggest that additional efforts may be required to support Batwa in locating Indigenous herbs outside of the forest, to recognize Indigenous healthcare systems as important in their own right and alongside biomedical systems, and to consider further integration of Indigenous and biomedical approaches to treatment in the district [27, 63].

## Implications for health policy, practice, and research

These research results on the linkages between the Batwa's displacement and their health-seeking behaviour for acute gastrointestinal illness present important recommendations for healthcare practice, public health, and health policy more broadly:

1. **Access to the forest.** To support Batwa place-connections–which intersect with all aspects of health and wellbeing–and to support continual multigenerational knowledge-sharing about Indigenous treatments, programs and policy may need to be enacted to permit Batwa access to the forest. Acknowledgement and respect for Batwa land rights, and, fundamentally, the right to self-determination, could have a significant impact on Batwa health and

health-seeking behaviour for acute gastrointestinal illness and more broadly [64, 65]. As stated by the WHO commission on social determinants of health, this may require "[restoration] of land necessary for sustaining Indigenous culture and livelihoods" [66:2] or Batwa Landback and returning to the forest, as discussed by Kokunda and colleagues [19].

2. **Continued advancement of community-based and community-led organizations.** To develop contextually-appropriate and effective policy or programmatic interventions, it is necessary to understand the scale, from local to global, at which "contextual processes"– structural determinants of health–operate, and at what scale their impacts are felt [7:1833]. For example, Batwa displacement operates at multiple scales, and by mutually-reinforcing processes: A national and international conservation initiative was enacted in a local place, and reinforced by law and policies governed at a national level [21, 23–25]. International, national, and local, grassroots organizations continue to work with cross-disciplinary goals to support Batwa health and livelihoods [20]. Continued coordination and facilitation of collaborative partnerships between these organizations–alongside consideration of the scale and mechanisms through which those partnerships can be most effective–may be beneficial in supporting Batwa health and wellbeing. Advancement of rights-based advocacy organization, the United Organization for Batwa Development in Uganda (UOBDU), or enhanced support for the Batwa Development Program, are examples of action to facilitate community health and development with, by, and for Batwa. The work of such organizations, strengthened through continual collaboration, may support Batwa connection to the land and livelihoods [65], and also advance supportive and accessible Indigenous and biomedical healthcare options for Batwa.

3. **Further research that examines people-place connections and access to locally-available, place-specific resources** [7]. The accessibility of local resources depends upon a complex interplay of sociocultural, cosmological, ontological, and epistemological processes, all of which relate Indigenous Peoples to place [1, 6, 56]. Future research may focus on understanding pathways through which Batwa health and wellbeing may be further supported in their new, agrarian context. Furthermore, researchers suggest that our own learning and research frameworks should be place-based and contextual–that is, inclusive of metrics and methods that scrutinize the interconnectedness of social, cultural, political, economic, and environmental phenomena and their mutually-reinforcing influence on health, wellbeing, and health-seeking behaviour [67, 68].

## Conclusion

This research reiterates that the Batwa's eviction from the forest continues to have wide-ranging consequences for Batwa health, wellbeing, and livelihoods [21–26]. The Batwa's forced transition to life outside the forest, their 'intellectual access' to healthcare, and Batwa identity and way of life, influence their health-seeking behaviour for acute gastrointestinal illness, a critical public health issue to be addressed. Further research may focus on place as a determinant of health [7, 53], not only for Batwa, but also for other communities with land-based livelihoods. For, as stated in the report from the Symposium on the Social Determinants of Indigenous Health, "another fundamental health determinant is the disruption or severance of ties of Indigenous Peoples to their land, weakening or destroying closely associated cultural practices and participation in the traditional economy essential for health and well being" [66:2]. For Batwa, such cultural practices in the forest and participation in a hunting and gathering economy were fundamental to health and wellbeing. To reduce the burden of acute

gastrointestinal illness, then, will require focused attention to Batwa health-seeking behaviour, and how timely care can be facilitated amid their new context and livelihoods.

## Supporting information

**S1 Checklist. Inclusivity in global research.**
(DOCX)

**S1 File. Health-seeking behaviour excerpt from semi-structured interview guide.**
(DOCX)

**S2 File. Health-seeking behaviour excerpt from focus group interview guide.**
(DOCX)

**S3 File. Additional, salient quotations on health-seeking behaviour for acute gastrointestinal illness in the context of Batwa connection to place and displacement.**
(DOCX)

## Acknowledgments

We express our gratitude to the participants from the communities of Bikuuto, Buhoma, Byumba, Karehe, Kebiremu, Kihembe, Kitahurira, Kitariro, Mukongoro, and Rulangara settlements in Uganda for their generous contributions to this research. We would also like to thank interviewees for their time and willingness to participate. Thank you to community researchers Evas Ninsiima and Yosam Besigensi, as well as local researchers Fortunate Twebaze and Grace Asasira, for their tireless work providing facilitation and interpretation during data collection. Grace has since passed away, and we remember her for her significant contributions to this work. The research reported here is part of a broader, international project entitled the "Indigenous Health Adaptation to Climate Change (IHACC)" project, with parallel field sites in the Canadian Arctic and Peru. Thank you to members of the IHACC Research Team for their support of and guidance in this work: Lea Berrang-Ford, Cesar Carcamo, Victoria Edge, James Ford, Sherilee Harper, Alejandro Llanos, Shuaib Lwasa, Didacus Namanya, and Carol Zavaleta.

## Author Contributions

**Conceptualization:** Lea Berrang-Ford, Sierra Nicole Clark, Kaitlin Patterson, Shuaib Lwasa, Didacus Namanya, Sherilee L. Harper.

**Formal analysis:** Laura Jane Brubacher.

**Funding acquisition:** Lea Berrang-Ford, Shuaib Lwasa, Didacus Namanya, Sherilee L. Harper.

**Investigation:** Sierra Nicole Clark, Kaitlin Patterson, Sabastian Twesigomwe.

**Methodology:** Lea Berrang-Ford, Sierra Nicole Clark, Kaitlin Patterson, Shuaib Lwasa, Didacus Namanya, Sabastian Twesigomwe, Sherilee L. Harper.

**Supervision:** Lea Berrang-Ford, Sherilee L. Harper.

**Writing – original draft:** Laura Jane Brubacher.

**Writing – review & editing:** Laura Jane Brubacher, Lea Berrang-Ford, Sierra Nicole Clark, Kaitlin Patterson, Shuaib Lwasa, Didacus Namanya, Sherilee L. Harper.

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
