## [Decision Letter · Decision Letter 0]

15 Feb 2024

PGPH-D-23-02190

Place, displacement, and health-seeking behaviour among the Ugandan Batwa: A qualitative study

Dear Dr. Brubacher,

Thank you for submitting your manuscript to PLOS Global Public Health. After careful consideration, we feel that it has merit but does not fully meet PLOS Global Public Health’s publication criteria as it currently stands. Therefore, we invite you to submit a revised version of the manuscript that addresses the points raised during the review process.

We look forward to receiving your revised manuscript.

Kind regards,

Alhelí Calderón-Villarreal

Academic Editor

Journal Requirements:

Additional Editor Comments (if provided):

Thank you for submitting this article addressing an important public health issue among Indigenous Peoples in rural Uganda. Forced displacement and land dispossession can impact people’s lives in several ways. This study particularly explored Batwa’s connection to place and their health-seeking behavior for acute gastrointestinal illness using qualitative methods.

In addition to reviewer’s comments, here are some questions and comments that would help improving the structure and discussion of this manuscript:

• Two Research Ethics Boards (at McGill University and University of Guelph) based in Canada reviewed this article. Why not IRB from a University based in Uganda?

• How local universities and/or researchers participated in the conduction of this research. Is any coauthor from or located in Uganda?

• I suggest checking for repetition in results and discussion sections. Also, the focus in ‘gastrointestinal illnesses’ seems to be secondary in a large proportion of the results/discussion.

• Lack of access to water, sanitation and hygiene services are very important risk factors for gastrointestinal illnesses. Did the authors explore access to these services or other environmental risk factors?

• As I understand from this study, Batwa’s People used to practice their own medicine until were forced to leave their land. However, in the discussion section of “Batwa identity and way of life, and ‘intellectual access’ to care”, ‘Indigenous forms of healthcare’ were described as ‘complements’ to biomedical systems and not as their own system. I suggest being careful assuming the hierargical relationship between both healthcare systems.

• International recommendations about environmental health and animal protection argue that Indigenous community’s participation in environmental conservation is key to ensure sustainability. About 80% of all biodiversity in the world is protected by Indigenous Communities. The conservative initiative to ensure protection of a declining mountain gorilla population is a key element for this study and the Batwa’s Community and requires more attention in the manuscript. I suggest incorporating more details in introduction and discussion about the role the World Bank/Global Environment Facility Trust Fund is playing in this process and the decision or the possibility to let this population come back to their land. Particularly, in the discussion section, the authors could incorporate structural determinants of health involved.

• I wonder if letting the Batwa’s People come back to their ancestral land is a recommendation that can potentially impact their health-seeking behavior for acute gastrointestinal illness. I suggest discussing this point. 

Reviewers' comments:

Reviewer's Responses to Questions

**Comments to the Author**

1. Does this manuscript meet PLOS Global Public Health’s publication criteria? Is the manuscript technically sound, and do the data support the conclusions? The manuscript must describe methodologically and ethically rigorous research with conclusions that are appropriately drawn based on the data presented.

Reviewer #1: Yes

2. Has the statistical analysis been performed appropriately and rigorously?

Reviewer #1: N/A

3. Have the authors made all data underlying the findings in their manuscript fully available (please refer to the Data Availability Statement at the start of the manuscript PDF file)?

Reviewer #1: Yes

4. Is the manuscript presented in an intelligible fashion and written in standard English?

Reviewer #1: Yes

5. Review Comments to the Author

Reviewer #1: The manuscript brings forth the challenges faced by the indigenous peoples, especially in the present context where numerous transitions are taking place globally including the areas predominantly inhabited by indigenous peoples. These transitions are affecting the indigenous communities, making them more marginalized from the mainstream society. Therefore, it is important to address the challenges faced by them for effective formulation of context-specific policy as pointed out by the authors of this manuscript. However, there are a few comments below that the authors need to address as deemed appropriate.

Abstract and Introduction-

1. The authors have missed sharing the information on the incidence rate or burden of acute gastrointestinal illness among the Batwa communities. Why is it a critical public health issue among the Batwa communities?

Methods-

2. Partner communities and research approach: Kanungu district includes 47 private or government-funded health facilities that range in the level of care they offer- This sentence is not clear.

3. Data collection and analysis: Please describe the socio-economic background of participants-especially those from the communities.

Results-

4. It is also important to add the perceptions of the Batwa communities on acute gastrointestinal illness- for instance, what is known locally in the community, their understanding of the illness and when they decide to seek care- their perceptions about the severity of the diseases- if the information on the same has been collected by the researchers.

5. It is important to have uniformity while writing the quotes of each participant. Participant age, gender, and educational status may be mentioned after the quotes.

6. PLOS authors have the option to publish the peer review history of their article (what does this mean?). If published, this will include your full peer review and any attached files.

**Do you want your identity to be public for this peer review?** For information about this choice, including consent withdrawal, please see our Privacy Policy.

Reviewer #1: **Yes: **Alacrity Muksor

---

## [Decision Letter · Decision Letter 1]

17 May 2024

Place, displacement, and health-seeking behaviour among the Ugandan Batwa: A qualitative study

PGPH-D-23-02190R1

Dear Dr. Brubacher,

We are pleased to inform you that your manuscript 'Place, displacement, and health-seeking behaviour among the Ugandan Batwa: A qualitative study' has been provisionally accepted for publication in PLOS Global Public Health.

Best regards,

Alhelí Calderón-Villarreal, MD, PhD, MPH

Academic Editor

Thank you for submitted a revised version of the manuscript. The reviewer and myself have agreed that you addressed the comments and the manuscript can be accepted. Congratulations.

Reviewer Comments (if any, and for reference):

Reviewer's Responses to Questions

**Comments to the Author**

1. If the authors have adequately addressed your comments raised in a previous round of review and you feel that this manuscript is now acceptable for publication, you may indicate that here to bypass the “Comments to the Author” section, enter your conflict of interest statement in the “Confidential to Editor” section, and submit your "Accept" recommendation.

Reviewer #1: All comments have been addressed

2. Does this manuscript meet PLOS Global Public Health’s publication criteria? Is the manuscript technically sound, and do the data support the conclusions? The manuscript must describe methodologically and ethically rigorous research with conclusions that are appropriately drawn based on the data presented.

Reviewer #1: Yes

3. Has the statistical analysis been performed appropriately and rigorously?

Reviewer #1: N/A

4. Have the authors made all data underlying the findings in their manuscript fully available (please refer to the Data Availability Statement at the start of the manuscript PDF file)?

Reviewer #1: Yes

5. Is the manuscript presented in an intelligible fashion and written in standard English?

Reviewer #1: Yes

6. Review Comments to the Author

Reviewer #1: All the comments have been addressed.

7. PLOS authors have the option to publish the peer review history of their article (what does this mean?). If published, this will include your full peer review and any attached files.

**Do you want your identity to be public for this peer review?** For information about this choice, including consent withdrawal, please see our Privacy Policy.

Reviewer #1: **Yes: **Alacrity Muksor
